

# Permeability and seismic velocity anisotropy across a ductile-brittle fault zone in crystalline rock

Quinn C. Wenning[1], Claudio Madonna[1], Antoine de Haller[2], and Jean-Pierre Burg[1]

[1]Department of Earth Sciences, Institute of Geology, ETH Zurich, Switzerland
[2]Department of Earth Sciences, University of Geneva, Switzerland

**Correspondence:** Quinn C. Wenning (quinn.wenning@erdw.ethz.ch)

**Abstract.** This study characterizes the elastic and fluid flow properties systematically across a ductile-brittle fault zone in crystalline rock at the Grimsel Test Site underground research laboratory. Anisotropic seismic velocities and permeability measured every 0.1 m in the 0.7 m across the transition zone from the host Grimsel granodiorite to the fault core show that foliation-parallel *p*- and *s*- wave velocities systematically increase from the host rock towards the fault core, while permeability

is reduced nearest to the fault core. The results suggest that although brittle deformation has persisted in the recent evolution, antecedent ductile fabric continues to control the matrix elastic and fluid flow properties outside the fault core. The juxtaposition of the ductile strain zone next to the brittle zone, which is bounded inside the two mylonitic fault cores, causes a significant elastic, mechanical, and fluid flow heterogeneity, which has important implications for crustal deformation and fluid flow, and exploitation and use of geothermal energy and geologic waste storage. The results illustrate how physical characteristics of

faults in crystalline rocks change in fault zones during the ductile to brittle transitions.

*Copyright statement.* TEXT

## 1   Introduction

Brittle faults and ductile shear zones and their associated damage and high strain zones have a localized yet influential impact on crustal mechanics and fluid flow (e.g., Sibson, 1994; Faulkner et al., 2010). Physical properties in and around the fault core

and damage zone in the brittle regime generally differ from the host rock by several orders of magnitude, asserting tremendous influence on fluid flow, deformation, earthquake rupture, and the development of economically exploitable resources. Less is known about the nature of the physical properties of ductile shear zones, and their role in crustal mechanics and fluid flow distribution once they are exhumed. Thus, understanding the nature of the geometrical distribution and temporal evolution of the properties associated with ductile shear zones and their transition into brittle faults and damage zones is integral to assess

crustal mechanics and fluid flow distribution.

Characterization of brittle faults and damage zones has received much attention (see review Faulkner et al., 2010). Previous studies from the laboratory (cm) to field outcrop (km) scale have developed into generalized models for the mechanical and




hydraulic behavior of fault zones (e.g., Chester and Logan, 1986; Caine et al., 1996; Faulkner et al., 2003, 2010). These models suggest that the fault zone consists of single or multiple high-strain cores surrounded by a damage zone where the physical properties are a function of the rock matrix, fracture density, and fault core. Brittle faults generally increase in fracture density in the damage zone towards the fault core (Vermilye and Scholz, 1998; Wilson et al., 2003; Mitchell and Faulkner,

2009), thereby increasing permeability and reducing elastic and mechanical strength from the intact rock towards the central fault core. Mitchell and Faulkner (2012) show that the microfracture density that enhances permeability around brittle faults scales with displacement of the fault. Laboratory experiments on Westerly granite indicate that increasing permeability due to microfracturing occurs regardless of the tectonic faulting regime (Faulkner and Armitage, 2013).

    In the ductile deformation regime, shear zones are understood to develop anisotropic properties due to mineral alignment of

anisotropic minerals in preferred elongation directions (shape preferred orientation or SPO) and/or alignment of the crystallographic axis (crystallographic preferred orientation or CPO) of minerals (Mainprice, 2007). The characteristics of ductile shear zones have been studied in terms of their anisotropic velocity structure to assess observations in middle to lower crustal seismic reflectivity (see review Almqvist and Mainprice, 2017). Rey et al. (1994) suggested that the physical properties in ductile shear zones should also be considered as transitional (i.e., the seismic velocities would grade into the ductile shear zone core). The

strength of ductile shear zones is typically studied in terms of viscous rheology (e.g., Sibson, 1983). However, these shear zones are often 'frozen in' and preserve their textural features that when exhumed behave with elastic and frictional failure criteria in the upper crust. In preserved ductile shear zones, mechanical and fluid flow properties have typically been studied separately. Géraud et al. (1995) studied the porosity and mineral structure across a mylonitic shear zone. Using empirical relationships between porosity and pore throat diameter, these authors were able to discern that the permeability decreases in the highest

strained sample.

    The models for elasticity and permeability through brittle and ductile shear zones have been mostly derived from outcrop examples (Faulkner et al., 2010, and references therein). To date, there have been limited systematic mechanical and fluid flow studies on boreholes that directly penetrate fault zones. Recent drilling through the Alpine Fault in New Zealand revealed how ductile mylonites have been exhumed, altering the rocks to a typical brittle fault damage zone in the vicinity to the fault

(Allen et al., 2017). Although much precaution is taken in outcrop studies, core material provides the opportunity to sample systematically into a fault zone eliminating issues of surface weathering and processes that may alter physical properties. Additionally, focus on the relationship between elasticity, mechanical strength, and permeability in the transition zone and core of faults has been inherently focused on brittle structures. However, with the vitality of meeting sustainable energy demands via geothermal energy and promoting safe geological waste disposal, the impact of preserved ductile structures in granitic rocks

on mechanics and fluid flow are also of increasing importance.

    Recent drilling at the Grimsel Test Site (GTS), an underground research laboratory in central Switzerland, penetrated a ∼5 m thick ductile-brittle fault damage zone relict in the Grimsel granodiorite host rock (Amann et al., 2017). Alpine tectonism produced multiple stages of ductile and subsequently brittle deformation in the Grimsel granodiorite, a member of the Aar massif (e.g., Rolland et al., 2009; Belgrano et al., 2016; Wehrens et al., 2016, 2017). The relict shear zone penetrated by the

borehole is bounded by two foliated shear zones, which initially localized ductile deformation and further reactivated brittlely



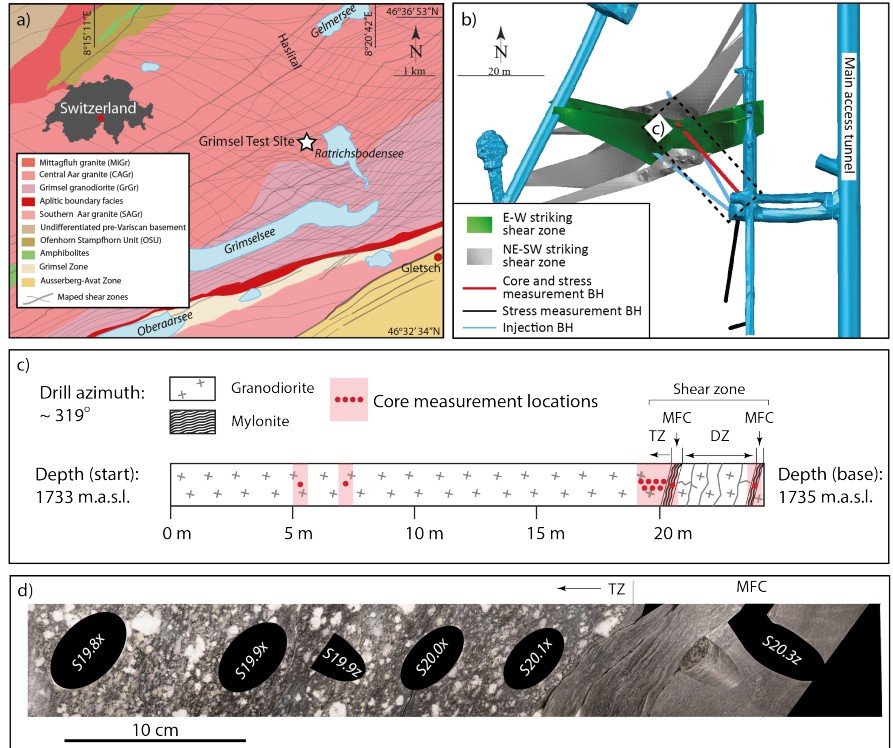

**Figure 1.** a) Geologic map of the Grimsel pass region (after Wehrens et al., 2016, 2017), b) borehole orientations (this study - red borehole) with respect to the underground research laboratory (after Krietsch et al., 2017; Amann et al., 2017), c) the borehole in this study depicting the location of the damage zone with coring and stress measurement locations projected along the borehole. The shear zone at the base is divided into three components: 1) transition zone (TZ), 2) mylonitic fault core (MFC), and 3) damage zone (DZ). d) Saw cut cross section through part of the transition zone and the mylonitic fault core (black ellipses show subcored cylinders inclined to the cut surface).

between the two foliated shear zones. This study concentrates on characterizing the elastic and fluid flow properties from the surrounding granodiorite rock mass through the ductile transition zone into the bounding foliated shear zone. Seismic $p$- and $s$-wave velocities ($V_p$ and $V_s$) and gas permeability ($k$) were measured on core samples in the laboratory. This paper emphasizes the mineralogical changes entering the shear zone influence changes in physical properties near the ductile-brittle damage zone.

5 The results also provide insight on the transient behavior of faults during the transition from ductile to brittle regimes through exhumation processes, and provide insight on their effect on economic exploitation of such shear zones in terms of geothermal energy or geological waste disposal.

## 2 Geologic setting and core details

The GTS is located in the Aar Massif in central Switzerland (Figure 1a). The underground laboratory is situated in Grimsel

10 granodiorite in the Haslital valley. Schaltegger and Corfu (1992) place the age of the Grimsel granodiorite at 299 ± 2 Ma.



From field relations and dating the Grimsel granodiorite has the same age as the Central Aar granite. These intrusions post-date Variscan collision, and there are no identified pre-Alpine structures.

Detailed deformation histories of the Grimsel region are available (Challandes et al., 2008; Rolland et al., 2009; Belgrano et al., 2016; Wehrens et al., 2016, 2017), which are summarized here. From argon-argon and rubidium-strontium dating, as well as field relations, beginning around 21 Ma and continuing until approximately 10 Ma, ductile deformation resulting from transpression created NNE-SSW, E-W, and NW-SE striking shear zones with steep dips to the south (Figure 1a). Ductile deformation is believed to occur in two stages: 1) from 21-17 Ma NNW-vergent thrusting and 2) from 14-10 Ma transpression causing dextral shearing of preferentially oriented oversteepened stage 1 structures. Beginning around 9 Ma steady exhumation caused retrograde ductile-brittle deformation in the form of discrete fractures, and subsequent embrittlement of these shear zones, which has produced fault breccias, cataclasites, and fault gouge.

The Aar granites experienced 300-450°C and 150 to 250 MPa peak conditions during Alpine metamorphism (Steck, 1968; Rolland et al., 2009; Wehrens et al., 2016, 2017). Our thin section observation shows fracturing of feldspar and undulose extinction along with subgrain boundaires in quartz, which are consistent with the inferred metamorphic temperatures.

In 2015, a series of boreholes were drilled in the Grimsel granodiorite (Figure 1b) for stress measurements, petrophysical property characterization, and hydraulic stimulation of the shear zones (Amann et al., 2017). The core material used in this study comes from the borehole drilled from an offset of the main tunnel in the GTS that penetrates two parallel shear zones. The well was drilled from 480 m below the ground surface in a subhorizontal trajectory with an azimuth of 319°. The well penetrates $\sim$ 20 m of mostly non-fractured granodiorite (Figure 1c). The granodiorite is foliated and at 20.2 m intersects a 20 cm thick foliated mylonitic shear zone, also defined as a foliated mylonitic fault core (MFC). The foliation intensity in the granodiorite decreases towards the host rock $\sim$ 0.5 m from the MFC through the transition zone (TZ) and is concordant with the foliation in the steeply dipping E-W oriented MFC (Krietsch et al., 2017). The TZ has a gradual decrease in grainsize of both matrix grains and the felsic clasts with more frequent mylonitic shear bands towards the MFC (Figure 1d and 2). The MFC itself is heterogeneously banded with mylonite and ultramylonite layers. A brittle damage zone (DZ) mixed with small <5 cm thick mylonitic shear zones is bounded between the MFC at 20.2 m and another 20 cm thick MFC at the end of the borehole. Less than 1 mm aperture quartz filled fractures intersect the MFCs originating from within the damage zone. However, these do not appear to penetrate entirely through the MFCs.

## 3 Methods

### 3.1 Sample selection, preparation, and characterization

In order to determine the spatial relationship of the physical properties in the shear zone a continuous set of samples was cored every 0.1 m in the transition zone from 19.6 m to the boarder of the first MFC at 20.1 m. Abundant fractures in the damage zone between the two MFCs prevented continuous coring. Two mutually perpendicular core samples, one parallel ($x_1$) and one perpendicular ($x_3$) to the Grimsel granodiorite foliation were taken to characterize the physical property and anisotropy changes as a gradient away from the fault core. Sampling farther than 19.5 m was not possible due to previously made overcoring stress





measurements (Figure 1c). In order to optimize the number of samples, the $x_1$ direction was taken $\sim 15°$ off axis from the lineation (Figure 1d). Foliation perpendicular samples could not be taken at 19.5 and 20.1 m because of breaks in the core. To characterize the MFC, parallel and perpendicular to foliation samples were taken at 20.2 and 23.6 m, respectively. The samples are extremely fissile, long enough to perform only permeability measurements (i.e., their length to diameter ratio is less than

1:1). Additionally, two sets of perpendicular samples were taken 5 and 7 m from the start of the borehole as a background Grimsel granodiorite reference. The $x_1$ and $x_3$ samples were bored out of the core using a diamond drill bit ( 2.54 cm inner diameter) with water as the cooling fluid. The 2.49 to 5.56 cm long samples were grinded and polished to craft parallel ends.

Thin sections were prepared directly from the ends of the samples and observed under optical microscopy. Quantitative mineral analysis was obtained at the University of Geneva using QEMSCAN® Quanta 650F, an automated scanning electron

microscope with mineral identification based on a combination of back-scattered electron values, energy-dispersive X-ray spectra, and X-ray count rates. High resolution mineralogical and petrographic maps were obtained with the QEMSCAN® at a scanning resolution of 5 $\mu$m, which measures the mineral coverage in percent area.

## 3.2  Density, porosity, and permeability measurements

Measurements of matrix volume and mass were performed after the samples were dried in an oven at 100°C for 24 hours for

the granodiorite samples and 40°C for the fragile MFC samples. The matrix volume was measured using a helium pycnometer (AccuPyc 1330, Micromeritics®). The dry mass was measured with a precision balance. The bulk rock density $\rho_{bulk}$ was calculated as the dry mass divided by the matrix volume of the sample. The porosity ($\phi$) of each cylindrical sample was calculated from the geometrical volume ($V_{tot}$) and the matrix volume ($V_m$) from the helium pycnometer $\phi = (V_{tot} - V_m)/V_{tot}$.

A hydrostatic pressure vessel was used to measure the gas permeability of each sample (detailed description of the apparatus

and measurement technique in Pini et al. (2009)). The hydrostatic pressure vessel is equipped to measure samples of 2.5 cm in diameter and up to $\sim$ 5 cm in length at confining pressures up to 20 MPa. Hydraulic oil is used as the confining fluid, which is controlled with a screw type displacement pump that regulates the confining pressure within $\pm$ 0.05 MPa. The sample assembly consists of the cylindrical rock specimen placed between two stainless steel disks fastened by a soft PVC tube to isolate the sample from the confining fluid. The two stainless steel disks have interconnected circular grooves to distribute

the fluid across the cross sectional area of the sample. The disks are connected via a plumbing system to the upstream and downstream reservoirs, which can be isolated and filled with the injected gas. The upstream and downstream reservoir, plus their associated plumbing network, have volumes of 50.8 cm$^3$ and 21.2 cm$^3$, respectively. The gas pressure in the two reservoirs is measured within 0.05 %.

Due to the low porosity and permeability in the granodioirite and MFC, the transient step technique was used to perform

and analyze the flow experiments (Brace et al., 1968). Experiments were performed at room temperature and an effective pressure of 10 MPa, chosen to represent the effective stress conditions in the GTS. Three confining pressure and pore pressure configurations that preserved an effective pressure of 10 MPa were performed to assess the so-called Klinkenberg gas slippage effect (Klinkenberg, 1941). For each sample a pressure difference of 0.5 MPa was imposed between the upstream ($P_{us}$) and downstream ($P_{ds}$) reservoir and allowed to equilibrate at each of the pore pressure configurations (e.g., $P_{us}$ and $P_{ds}$ = 1.0 and





0.5 MPa, 3.0 and 2.5 MPa, and 7.0 and 6.5 MPa, respectively). In some cases, the sample was so impermeable that reaching a full equilibrium between the up and down stream reservoir was not possible within laboratory timescale. For these samples, only the beginning part of the partial pressure gradient equilibration has been assessed.

Hsieh et al. (1981) developed a full analytical solution to the differential equation describing the gas pressure inside the sample as a function of the distance along the sample and time to estimate permeability. Dicker and Smits (1988) developed a simple analytical expression to estimate permeability from the measured pressure curves. The simple analytical solution:

$$k = \frac{\beta\mu\phi L^2 s}{f(V_{sa}/V_{us}, V_{sa}/V_{ds})} \tag{1}$$

is a function of the compressibility, $\beta$, and viscosity, $\mu$, porosity, $\phi$, length of the sample, $L$, slope of the differential pressure vs. time, $s$, and a function of the ratio between the volume of the sample ($V_{sa}$) and the volume of the up and downstream reservoirs ($V_{us}$ and $V_{ds}$, respectively). The solution is accurate within 0.3 % of the full experession if the pore volume is less than the reservoir volumes, which is true for our experiments. Since the pressure difference in the two reservoirs is small we used an average pore pressure to determine the compressibility and viscosity of the argon gas using the NIST database (NIST, 2017).

### 3.3 Elastic wave velocity measurements and calculations

A separate hydrostatic oil-medium pressure vessel, capable of reaching high confining pressures, was used to measure the p- and s- ultrasonic elastic wave velocities using the pulse transmission technique (Birch, 1960). The measurements were conducted on the mutually perpendicular samples up to 260 MPa and at room temperature conditions (detailed description of the measurements described in Zappone et al. (2000)). The mechanical impulse is directed into the sample by mounting the lead zirconate titanate piezoceramic transducer transducer inside a 'head' assembly that also contains a buffer rod, reducing the dispersion of energy. The setup is configured so that one transducer transforms the electrical impulse (1 MHz resonance frequency) and emits a mechanical wave at the coupling of the transducer with the sample. After passing through the sample, another transducer converts the mechanical wave back into an electrical signal. The electronic system consists of a Hewlett-Packard® 214B Pulse Generator that is connected to the transducers with coaxial cables and the output is recorded directly with a computer. To prevent oil seepage from the confining fluid into the sample, a thin polyolefin heat shrink tube is fitted over the ends of the transducers and the sample.

The velocity in the rock is given by

$$V_{p,s} = \frac{L}{t_{sample}} \quad \text{with} \quad t_{rock} = t_{total} - t_{system} \tag{2}$$

where the p- and s- wave velocities, $V_{p,s}$, are a function of the travel time through the sample, $t_{sample}$, and its length, $L$. The travel time through the sample is determined by subtracting the travel time of the cabling in the source/receiver system, $t_{system}$, from the total time of flight of the impulses recorded, $t_{total}$.

The waveforms are recorded at stepwise increases or decreases in pressure in the loading and unloading cycles performed for each p- and s-wave experiment. Measurements were recorded across the full pressure range of 30 to 260 MPa to investigate





the properties closest to present day low pressure conditions at the GTS (Minimum principal stress 8 to 12 MPa, maximum principal stress 13-17 MPa Krietsch et al. (2017)) and to study the poro-elastic effect on seismic velocities after crack closure at high pressure (Birch, 1960). The measurements were made at room temperature and dry, undrained conditions. Recordings of the wave form were measured within $\pm 2$ MPa and a travel time accuracy of $\pm 0.01$ $\mu$s.

Velocity anisotropy (A$V$) was estimated from the maximum, minimum, and mean velocities using

$$AV_{p,s} = \frac{V_{p,s\,max} - V_{p,s\,min}}{V_{p,s\,mean}} * 100 \tag{3}$$

Estimates of the dynamic elastic moduli were also calculated for each experiment. The $p$- and $s$-wave moduli are represented in the general form as $c_{xx} = \rho V_{p,s}^2$. The $p$-wave moduli for the vertical ($x_3$) and maximum horizontal ($x_1$) samples are represented by $c_{33}$ and $c_{11}$, respectively. Similarly, the $s$-wave moduli, also known as shear modulus ($\mu$), for the vertical and maximum horizontal samples are represented by $c_{44}$ and $c_{66}$, respectively. The elastic moduli are estimated by applying the isotropic

equations to the vertical and horizontal components separately in order to estimate the $p$- and $s$- wave moduli (Mavko et al., 2009). Sone and Zoback (2013) show the error in applying the isotropic equations to the vertical and horizontal components separately in the absence of having the 45°-oriented sample is negligible. The dynamic Young's moduli are approximated for the parallel ($E_1$) and perpendicular ($E_3$) components using the following equations:

$$E_1 = \frac{c_{66}(3c_{11} - 4c_{66})}{c_{11} - c_{66}} \tag{4}$$

$$E_3 = \frac{c_{44}(3c_{33} - 4c_{44})}{c_{33} - c_{44}} \tag{5}$$

The dynamic Poisson's ratio for the parallel ($\nu_1$) and perpendicular ($\nu_3$) sample is calculated using the isotropic equation

$$\nu = \frac{1}{2}\frac{(V_p/V_s)^2 - 2}{[(V_p/V_s)^2 - 1]} \tag{6}$$

The dynamic bulk modulus for the parallel ($K_1$) and perpendicular ($K_3$) sample is calculated using the isotropic equation

$$K = \rho(V_p^2 - \frac{4}{3}V_s^2) \tag{7}$$

## 4 Results

### 4.1 Characterization

In general there is a decrease in grain size in the TZ toward the MFC (Figure 1d). Additionally, mm thick shear bands become more frequent nearer to the MFC until reaching the sharp boundary with the MFC. The MFC itself is heterogeneously layered

and folded. The compositional and microstructural transition from the 'host' granodiorite, through the transition zone (TZ), and the mylonitic fault core (MFC) are depicted in Figure 2 and the rock composition is summarized in Table 1. The density



**Table 1.** Summary of sample composition: Sample name refers to depth in the borehole, rock type refers to either the host granodiorite, transition zone (TZ), and mylonitic fault core (MFC), dry bulk density and porosity is reported as an average of individual measurements for each sample ($x_1$ and $x_3$), and mineral composition is derived from the QEMSCAN analysis of the $x_1$-thin section in % area. Mineral abbreviations: Bt = biotite, Phl = phlogopite, Ms = muscovite, Ep = epidote, Ab = albite, Kfs = K-feldspar, and Qz = quartz.

| Sample | Rock type | Density [g/cm$^2$] | Porosity [%] | Bt+Phl [%] | Ms [%] | Ep [%] | Ab [%] | Kfs [%] | Qz [%] | Other [%] |
|--------|-----------|--------------------|--------------|------------|--------|--------|--------|---------|--------|-----------|
| S5 | Host | 2.73 | <1 | 9 | 6 | 6 | 43 | 17 | 17 | 2 |
| S7 | Host | 2.73 | <1 | 10 | 4 | 5 | 39 | 15 | 26 | 2 |
| S19.5 | TZ | 2.74 | <1 | 9 | 9 | 3 | 46 | 5 | 25 | 2 |
| S19.6 | TZ | 2.75 | <1 | 6 | 8 | 5 | 45 | 5 | 28 | 2 |
| S19.7 | TZ | 2.76 | <1 | 10 | 7 | 7 | 50 | 3 | 21 | 2 |
| S19.8 | TZ | 2.75 | <1 | 9 | 7 | 4 | 45 | 3 | 30 | 2 |
| S19.9 | TZ | 2.73 | <1 | 12 | 10 | 1 | 56 | 3 | 16 | 2 |
| S20.0 | TZ | 2.77 | <1 | 15 | 13 | 1 | 42 | 4 | 22 | 2 |
| S20.1 | TZ | 2.73 | <1 | 13 | 16 | 0 | 42 | 3 | 25 | 2 |
| S23.6 | MFC | 2.82 | <1 | 27 | 0 | 12 | 31 | 5 | 22 | 2 |

of granodiorite samples irrespective of their proximity to the MFC varies between 2.72 and 2.78 g/cm$^3$ with porosity varying between 0.4 to 1% (Table 1). In general, the samples are void of open microcracks, thus the porosity occurs between grain contacts (i.e., intergranular micropores). The density of the MFC from both sampling locations is 2.80 and 2.84 g/cm$^3$ and porosity estimates are 0 and 1%, respectively.

The samples (S5 to S20.1) from the granodiorite are made up of various amounts of plagioclase (albite), quartz, K-feldspar, biotite/phlogopite, muscovite, and epidote. The amount of each mineral phase and microstructure depends on the vicinity to the mylonitic fault core. In the samples taken from the 'host' granodiorite (S5 and S7) as well as samples farthest from the fault core (S19.5 and S19.6) the microstructure and composition is similar. Plagioclase is the most abundant mineral phase ($\sim$ 40%). The sub mm to >10 mm big plagioclase grains are rounded to subangular. The grain size of plagioclase varies

from sub-mm to >10 mm. Needle like sericite inclusions (<0.1 mm) form within the plagioclase cleavage planes, and indicate hydrothermal alteration occurred. Quartz subgrains also develops along the boundaries and within large plagioclase grains. In larger plagioclase grains brittle fractures are filled with biotite and quartz. Quartz is the second most abundant mineral phase ($\sim$ 17 to 25%). Quartz grains of variable size (<1-2 mm) typically occur as many rounded to subhedral individual subgrains that form lenses or develop in the strain shadows of plagioclase clasts (Figure 2). The main difference between the 'host'

granodiorite (S5 and S7) and the beginning of the transition zone (S19.5 and S19.6) is the K-feldspar concentration, which is $\sim$ 15 to 17% and $\sim$ 5%, respectively. Phyllosilicates in the form of biotite and muscovite form anastamosing lenses of mixed muscovite and biotite with variable thickness across the thin section, which comprise about 15 to 18% of the total mineralogy. Biotite forms <0.1 to 1 mm grains, of which the individual grains are randomly oriented in the anastamosing lenses.



A progressive change in the overall microstructure, state of the individual minerals, and the mineral composition is observed in the transition zone between samples S19.7 and S20.1. In Sample S19.7 the foliation becomes more continuous across the thin section when compared to the 'host' granodiorite samples. Plagioclase deforms brittlely in the form of fracturing (Figure 2), while grain boundary migration, undulose extinction, and subgrain rotation is observed in the quartz grains indicates ductility.

Plagioclase and quartz grain size are similar to 'host' granodiorite. Lenses of biotite and muscovite extend across the thin section more continuously, however the lenses form variable thicknesses that wrap around the intermixed plagioclase and quartz. The anastomosing lenses are still present. In the samples nearest the MFC (S20.0 and S20.1) the continuity and thickness of the mica-rich layers across the sample are the most developed. The foliation planes orient with the parallel alignment of the individual grains. Biotite and muscovite grains are especially larger in samples S20.0 and S20.1, where individual grains

can be >5 mm long. While lenses of fine grained phyllosilicates occur, the overall grain size, continuity, layer thickness, and orientation of the individual grains is greater and more continuous. The total phyllosilicate amount increases from $\sim$ 15-18% in samples S5 to S19.6 to $\sim$ 30% in samples S20.0 and S20.1, with the other tectosilicates (plagioclase, K-feldspar, and quartz) reducing as a result. Biotite and quartz appear in the strain shadows of the plagioclase clasts (Figure 2).

The MFC sample (S23.6) is constituted of very fine grained unltramylonitic (more than 90% grain size reduction) plagioclase

(31%), biotite (27%), quartz (22%), and epidote (12%) making up the main mineral constituents. The foliation is defined by the biotite-quartz/plagioclase preferred shape orientation (layers typically <0.1 mm). Recrystallized plagioclase and quartz form rotated clasts within the biotite-quartz foliation. The shear zones in the region are often interpreted as former mafic dykes (e.g., Wehrens et al., 2017). The MFC has a gradient of deformation between the Grimsel granodiorite in the TZ and the MFC, and most notably there is a heterogeneous layering within the ultramylonite with larger grain lenses that are compositionally similar

to the granodiorite (e.g., Figure 1d). We interpret this structure more broadly as a mylonitic shear zone with a strain gradient of decreasing deformation away from the fault cores.

## 4.2   Velocity measurements and elastic moduli calculation

Seismic velocities (Figure 3 and Tabel 2) are reported for the 30 MPa confining pressure measurement (i.e., the closest measurement to the stress magnitudes in the GTS). The measured velocities parallel to foliation at 30 MPa, (Figure 3 and Tabel 2)

show an increase across the $\sim$ 0.5 m transition zone. In the two samples taken from the 'host' granodiorite (S5 and S7) $p$-wave velocity parallel to the foliation ($V_{px_1}$) are $\sim$ 5.5 km/s. Samples S19.5 to S19.7 taken from the beginning of the transition zone have comparable velocities to the 'host' granodiorite samples (5.55 to 5.61 km/s). Transitioning towards the fault core the $V_{px_1}$ increases steadily and reaches a maximum in sample S20.1 (6.14 km/s) directly adjacent to the MFC at 20.2 m. The $s$-wave velocity follows a similar trend where the host samples and the samples farthest from the fault core have a $V_{sx_1}$ of $\sim$

3.42 to 3.54 km/s and the velocities increase steadily and reach a maximum nearest the MFC (S20.1: $V_{sx_1}$ = 3.83 km/s). The velocities measured perpendicular to the foliation, $V_{px_3}$ and $V_{sx_3}$ fluctuate without a consistent trend from 4.98 to 5.21 km/s and 3.20 to 3.35 km/s, respectively. The $p$- and $s$-wave anisotropy is generally lower away from and higher near the MFC. However, there are outliers (e.g., S19.6), which can be attributed to bias from either a slower $x_3$ velocity or a faster $x_1$ velocity.





The seismic velocities are in general agreement with previous measurements on Grimsel granodiorite (Keusen et al., 1989) and other granodiorite samples (Jones and Nur, 1982).

The dynamic elastic moduli behave like the velocities because the density remains consistent for each sample. Therefore, the velocities exert greater influence on the $x_1$ and $x_3$ moduli. Approaching the MFC each respective dynamic elastic moduli

increases by 10 to 20 GPa for the $x_1$ sample, while the $x_3$ remains almost constant, which corroborates previous measurements on Grimsel granodiorite (Keusen et al., 1989). The dynamic Poisson's ratio remains relatively uniform throughout.

Additonally, seismic velocities were measured up to confining pressures of 260 MPa in order to determine the intrinsic crack free velocities of the rocks (Figure 4). Both $V_p$ and $V_s$ velocity contours show that for a given confining pressure, the velocities parallel to foliation tend to increase to maximum values closest to the fault core and that there is minimal sporadic variation in

the perpendicular to foliation velocity measurements.

### 4.3    Permeability measurements

Permeability decreases (Figure 3 and Table 2) from the host granodiorite and farthest samples in the transition zone (0.99 to $8.38 \times 10^{-19}$ m$^2$) towards the samples nearest the fault core (0.03 to $1.89 \times 10^{-19}$ m$^2$) along the $x_1$ direction. The permeability perpendicular to the foliation $x_3$ fluctuates from 0.52 to $4.14 \times 10^{-19}$ m$^2$. Permeabilities of similar host Grimsel granodiorite

at 5 MPa are $10^{-18}$ to $10^{-20}$ m$^2$ (David and Wassermann, 2017) and measurements on Kola granodiorite samples range from approximately $10^{-18}$ to $10^{-20}$ m$^2$ at effective pressures of 10 to 50 MPa (Morrow et al., 1994). Directional permeability of the mylonitic fault core was measured on two samples from separate foliated shear zones due to difficulties in sample preparation (i.e., $x_3$ is from 20.3 m depth and $x_1$ is from 23.6 m depth). The permeability is $0.03 \times 10^{-19}$ m$^2$ parallel to foliation and $0.57 \times 10^{-19}$ m$^2$ perpendicular to the foliation. Flow along the boundary of a quartz-filled vein that cross-cuts the perpendicular

sample is believed to cause the increase in permeability perpendicular to foliation in the mylonitic fault core.



**Table 2.** Summary of density, porosity, elastic properties and anisotropy, and permeability obtained from laboratory measurements. Laboratory velocities measured at 30 MPa confining pressure and permeability measured at 10 MPa effective pressure.

| Sample | Direction | $V_p$ [km/s] | mean $V_p$ [km/s] | $AV_p$ [%] | $V_s$ [km/s] | mean $V_s$ [km/s] | $AV_s$ [%] | $E$ [GPa] | $\mu$ [GPa] | $K$ [GPa] | $\nu$ | $k$ [m$^2$] $\times10^{-19}$ |
|---|---|---|---|---|---|---|---|---|---|---|---|---|
| SBH5 | $x_1$ | 5.52 | 5.34 | 6.75 | 3.43 | 3.35 | 4.52 | 76 | 32 | 40 | 0.19 | 8.38 |
| | $x_3$ | 5.16 | | | 3.28 | | | 68 | 29 | 33 | 0.16 | 4.14 |
| SBH7 | $x_1$ | 5.50 | 5.30 | 7.55 | 3.42 | 3.32 | 6.30 | 76 | 32 | 40 | 0.18 | 5.95 |
| | $x_3$ | 5.10 | | | 3.21 | | | 66 | 28 | 33 | 0.17 | 1.70 |
| SBH19.5 | $x_1$ | 5.59 | - | - | 3.49 | - | - | 79 | 33 | 41 | 0.18 | 0.99 |
| | $x_3$ | - | | | - | | | - | - | - | - | - |
| SBH19.6 | $x_1$ | 5.61 | 5.30 | 11.55 | 3.54 | 3.37 | 10.01 | 81 | 35 | 41 | 0.17 | 2.11 |
| | $x_3$ | 4.99 | | | 3.20 | | | 65 | 28 | 31 | 0.15 | 2.72 |
| SBH19.7 | $x_1$ | 5.55 | 5.26 | 10.79 | 3.51 | 3.43 | 4.75 | 79 | 34 | 39 | 0.16 | 5.29 |
| | $x_3$ | 4.98 | | | 3.35 | | | 67 | 31 | 27 | 0.09 | 1.36 |
| SBH19.8 | $x_1$ | 5.70 | 5.37 | 12.14 | 3.55 | 3.43 | 7.15 | 82 | 35 | 43 | 0.18 | 5.60 |
| | $x_3$ | 5.05 | | | 3.31 | | | 68 | 30 | 30 | 0.12 | 1.05 |
| SBH19.9 | $x_1$ | 5.76 | 5.38 | 14.43 | 3.63 | 3.48 | 8.28 | 84 | 36 | 43 | 0.17 | 1.89 |
| | $x_3$ | 4.99 | | | 3.34 | | | 67 | 30 | 27 | 0.09 | 1.37 |
| SBH20.0 | $x_1$ | 5.99 | 5.60 | 13.85 | 3.68 | 3.48 | 11.55 | 90 | 38 | 49 | 0.20 | 1.55 |
| | $x_3$ | 5.21 | | | 3.28 | | | 70 | 30 | 35 | 0.17 | 0.52 |
| SBH20.1 | $x_1$ | 6.14 | - | - | 3.83 | - | - | 95 | 40 | 50 | 0.18 | 0.03 |
| | $x_3$ | - | | | - | | | - | - | - | - | - |
| SBH20.3 | $x_3$ | - | | | - | | | - | - | - | - | 0.57 |
| SBH23.6 | $x_1$ | - | | | - | | | - | - | - | - | 0.03 |



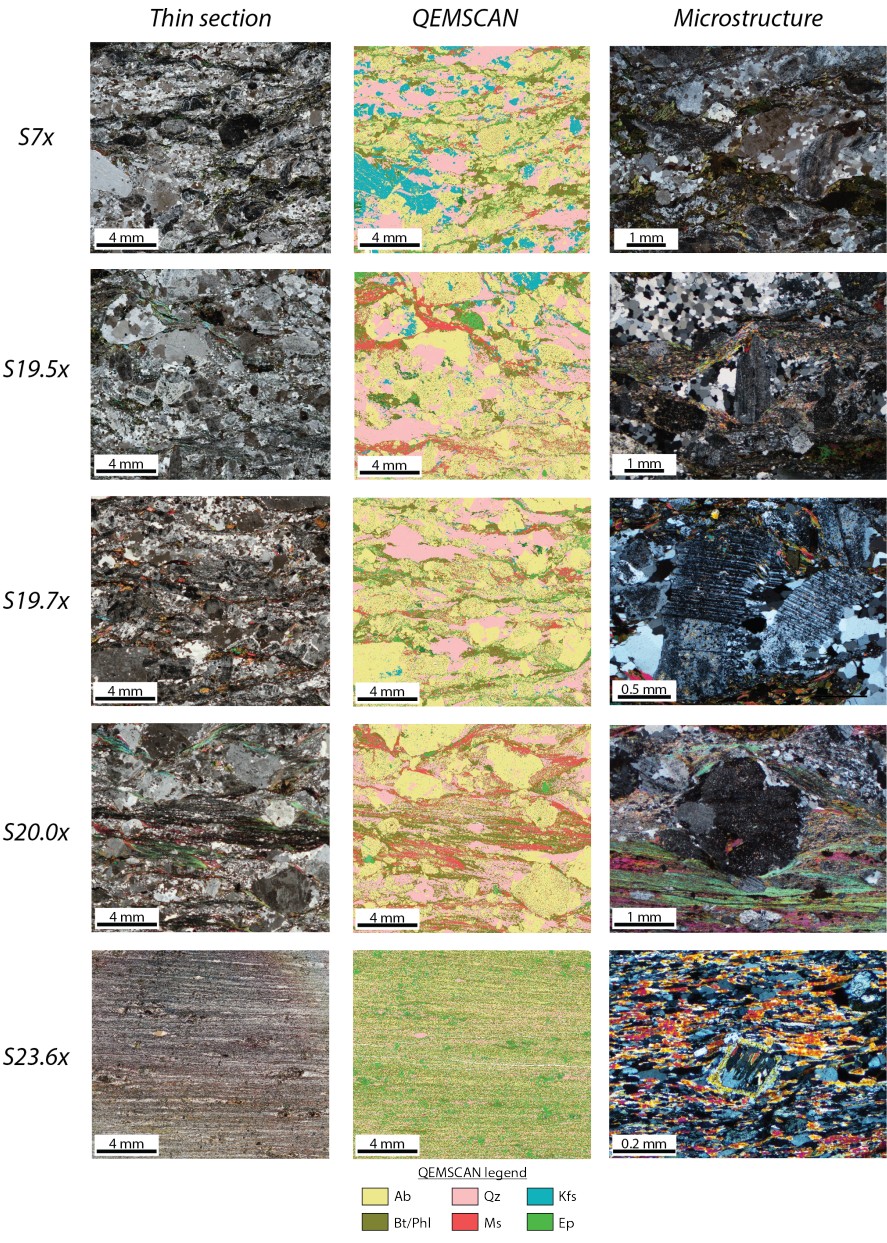

**Figure 2.** Optical microscopy and QEMSCAN® images of thin sections for each $x_1$ sample (S7x, S19.5x, S19.7x, S20.0x, and S23.6x). The left column shows the cross polarized light image of the sample with the corresponding QEMSCAN® area in the middle column. The right column depicts a particular microstructure of the sample. S7x - typical host rock texture with twinned feldspar and quartz. S19.5x - elongated feldspar with quartz in strain shadows between fine grained mica. S19.7x - fractured feldspar with quartz grains filling the fracture. S20.0x - Rounded sericitic feldspar clast with fine grained quartz and mica infilling the strain shadows between lenses of large mica grains and fine recrystallized quartz/mica. S23.6x - very fine grained foliated biotite and quartz grains surrounding euhedral plagioclase. Sample locations depicted in Figure 1.





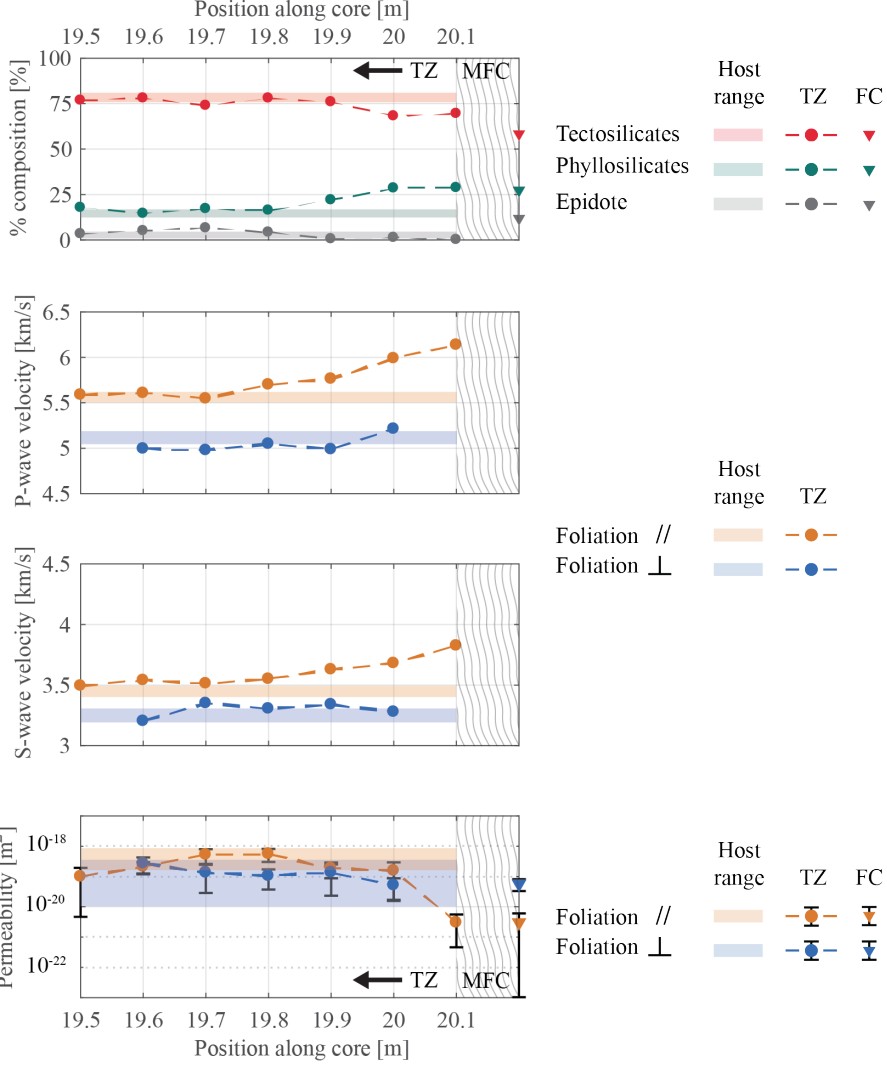

**Figure 3.** Sample composition, seismic velocity, and permeability across the transition zone into the fault core. Top panel: tectosilicates - quartz, plagioclase, and K-feldspar (red), phyllosilicates - biotite and muscovite (green), and epidote (grey). Middle panels: *p*- and *s*-wave velocity parallel to foliation (orange) and perpendicular to folication (blue). Bottom panel: permeability parallel to foliation (orange) and perpendicular (blue). Permeability is reported as the median value measured for each sample, and the bars show the range of values in terms of minimum and maximum. For all panels the shaded region in depicts the range of values from the host granodiorite samples (S5 and S7), the circular markers show the values measured through the transition zone (TZ), and the triangular markers show the values measured in the fault core (MFC). Error bars are depicted where error is larger than the marker size.





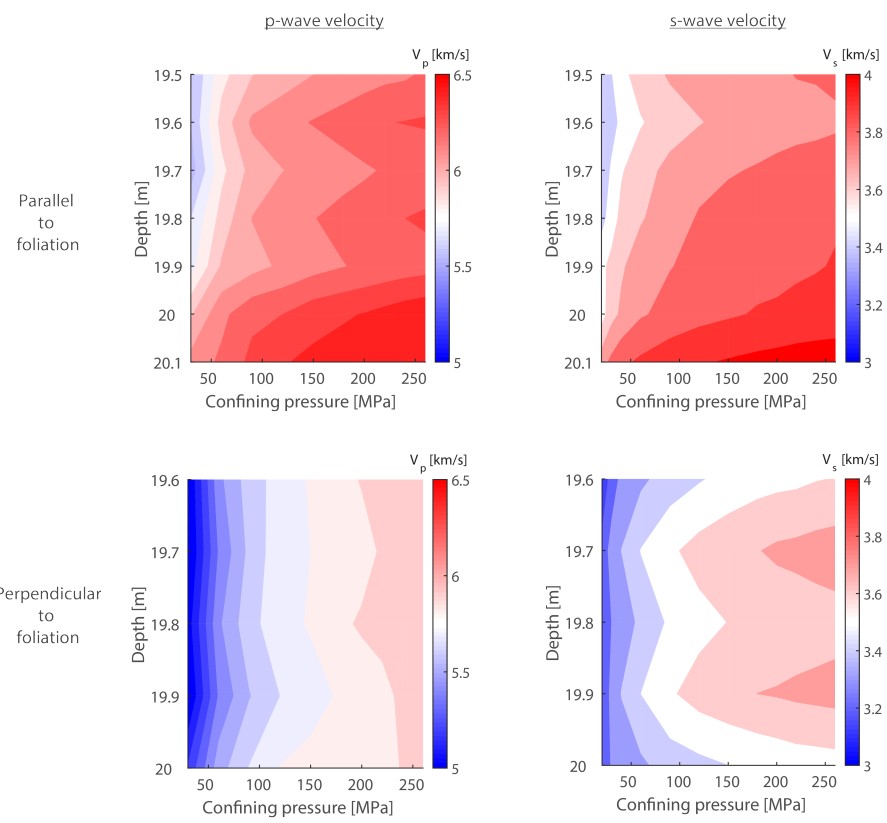

**Figure 4.** Contour plots showing the influence of confining pressure on seismic velocities along the core in the transition zone. Top panel displays the results for foliation parallel velocities ($x_1$) separated into $V_p$ (left) and $V_s$ (right). Bottom panel displays the results for foliation perpendicular velocities ($x_3$) separated into $V_p$ (left) and $V_s$ (right). For $V_p$ and $V_s$ red colors indicate faster velocities and blue colors depict slower velocities (magnitude defined on the colorbar).





## 5 Discussion

### 5.1 Shear zone characterization

Many studies on the transition of elastic and fluid flow properties in and around fault cores and damage zones have been concentrated on outcrop material of brittle faults (Faulkner et al., 2010). Caine et al. (1996) present models for fault core

geometries, with fault cores composed of fault gouge or cataclasite. Faulkner et al. (2010) expand this model to include both single fault core damage zones and damage zones made up of several anastamosing faults. Laboratory measurements on samples from natural fault systems have led to the development of brittle fault permeability and elastic or mechanical properties that are microfracture dependent. In such systems, damage is concentrated in the fault core, which produces fault gouge or cataclasite that can either be higher or lower in permeability than the surrounding host rock. In the host rock directly

contacting the fault zone, microfracturing due to strain displacement around the fault leads to increased permeability and decreased elastic or mechanical strength (Faulkner et al., 2006). Permeability decreases and elastic or mechanical strength increases moving away from the damage zone core, as the microfracture intensity decreases away from the shear zone (e.g., Vermilye and Scholz, 1998; Wilson et al., 2003; Mitchell and Faulkner, 2009).

In ductile shear zones the alignment of anisotropic minerals in CPO or SPO due to strain accumulation has been a central

focus for crustal reflectivity (e.g., Fountain et al., 1984; Jones and Nur, 1984; Kern and Wenk, 1990). Rey et al. (1994) discuss the existence of transition zones in which the strain progressively increases towards the ductile shear zone core, causing gradual changes in the physical properties around such faults. The permeability across a mylonitic ductile shear zone was estimated from relationships between porosity and pore throat radius (Géraud et al., 1995). They show that permeability is reduced in the central shear zone core, but is higher in the surrounding strain gradient, which is higher than the host rock. For shear zones

that have undergone the transition from ductile to brittle deformation a competing process between microfracture and mineral orientation controlled physical properties can be envisaged.

The shear zone selected for measurements of seismic velocities and permeability in this study preserve both ductile and overprinting brittle structures. The shear zone penetrated by the borehole at GTS is characterized by foliation aligned with the mylonitic fault core that developed under viscous flow deformation conditions (Wehrens et al., 2016). The foliation intensity

is highest nearest the mylonitic fault core and decreases into the host granodiorite. The brittle fractures, which are bounded between the mylonitic fault cores, formed during a later brittle overprint. Figure 3 shows that the elastic and permeability transition into the fault core is dissimilar to models derived from brittle fault zones, even though brittle deformation is evident in the damage zone. The measurements from GTS show a trend increasing velocities and stable to decreasing permeability in the plane parallel to foliation in the transition zone. In the direction perpendicular to foliation the velocities and permeability

have minor fluctuation in the vicinity to the fault core. The ductile strain gradient in the transition zone does not appear to be influenced by the later stages of brittle deformation, as indicated by the increased seismic velocities and slightly decreasing permeability parallel to foliation towards the core in the transition zone. Should there be a brittle overprint, velocities would be expected to decrease due to microfracturing, however this is not the case.




Instead, within the transition zone the both elastic and fluid flow properties are controlled by mineralogical changes in the rocks. Microfractures in thin section are scarce, thus most of the < 1 % porosity are intergranular micropores. It is important to note that these changes are localized within ∼ 1 m of the ductile fault core. Since the material is bored from an underground research lab, alteration processes and weathering should be suppressed in such samples. The mineralogy of the samples shows a gradual change in composition, loosing tectosilicates (Pl, Fsp, and Qz) and gaining phyllosilicates (Bt and Ms), through the transition zone (Figure 3). There is an increase in foliation intensity towards the fault core. The faster foliation-parallel velocities are controlled by the alignment of the platy phyllosilicate minerals (Lloyd et al., 2011b, a). Higher foliation-parallel permeability compared to flow perpendicular, as measured in the GTS samples, has been measured in previous studies (e.g., Faulkner and Rutter, 1998; Leclère et al., 2015; Wibberley and Shimamoto, 2003; Uehara and Shimamoto, 2004). In low grade to ductile deformation changes in the mineralogy and foliation structure alters the connection of intergranular micropores of the platy phyllosilicate and tectosilicate minerals (e.g., Faulkner and Rutter, 1998; Leclère et al., 2015; Géraud et al., 1995). In this study the changes in mineralogy, most notably the phyllosilicate to tectosilicate ratio, is a driver in both the velocity and permeability anisotropy, where microcracks do not have a driving role due to their scarcity.

Outside the MFCs, fractures along the borehole wall are uncommon, as indicated by optical televiewer images (Krietsch et al., 2017). Between the MFCs the density of fractures is high enough to have been termed damage zone. Although some fractures penetrate the MFCs, they are typically quartz-filled and generally do not connect the granodiorite on either side of the fault core. In the damage zone itself, fluid flow properties and elasticity are governed by the micro and macroscopic fractures. In the damage zone the velocities are decreased and the permeability increases indicated by logging and pump tests in the borehole (Jalali et al., 2017). Since the microfracturing does not appear to have influence outside the MFCs, displacement of these brittle features is likely small (Mitchell and Faulkner, 2012). Similar mapped faults in the region have cataclastic gouge or fault breccia (Belgrano et al., 2016), indicating that the fault at the GTS in not mature and has not accommodated much of the brittle displacement since the ductile structure is still preserved.

## 5.2 Ductile-brittle transition in the fault zone

The measurements at the GTS leads us to hypothesize how fault properties might vary not only in geometry, but also in the transient evolution of the fault itself. For the transition from a ductile to brittle fault system in crystalline rock, two end member behaviors can be envisaged, ductile and brittle. While the rock is undergoing ductile deformation and localizing along the mylonitic fault core shear zones the deformation processes would be accommodated by crystal-plastic flow. The highly strained and extreme recrystallization in the ultramylonite in the fault core, along with the seritization of the plagioclase in the transition zone indicate that fluids were present and likely localized in the fault core during deformation. However, once the deformation and ductile structures were frozen in, the ductile transition zone behaves in a manner where elasticity parallel to foliation increases transitioning from the host rock to the ductile core parallel to the foliation, and permeability decreases in the core. On the other hand in the brittle damage zone model, microfracturing induces permeability enhancement and weak elasticity nearest the fault core (e.g., Caine et al., 1996; Faulkner et al., 2010). This case study from the GTS is a hybrid between the two end member systems.



During the two ductile deformation phases slip was accommodated along localized foliated shear zones that are mylonitic and ultramylonitic (Challandes et al., 2008; Rolland et al., 2009; Belgrano et al., 2016; Wehrens et al., 2016, 2017). The transition from highly foliated and extremely recrystallized fault cores towards the host granodiorite represents a strain gradient, which is ∼ 0.5 m thick in this study (Figure 5a). The highest foliation intensity in the granodiorite nearest the fault core also

creates a change in bulk mineralogy (i.e., more phyllosilicates and less tectosilicates, Figure 3) and microstructure (i.e., more laminated, Figure 1d and 2), which alter the petrophysical properties that once 'frozen in' behave as those measured in the transition zone and mylonitic fault core in this study (i.e., higher seismic velocity parallel to foliation and lower permeability nearest the fault core). Fluid flow channelization in ductile shear zones have been argued based on mobile elements (Ca, Mg, Na, and K) concentration, stabile isotopes ($\delta^{18}O$), and fluid phase observations (Etheridge et al., 1983; Marquer and Burkhard,

1992). However, in this study and the study by Géraud et al. (1995) the lowest permeability measurements come from the fault core. While the current measurements come from the 'frozen in' ductile microstructure, dynamic porosity changes might be occurring during deformation (e.g., Mancktelow et al., 1998), which could enhance the permeability in the fault core. It is possible that the ductile shear zone would behave in such a way that the long term permeability of the shear zone is low, creating a pressure seal. Then during rupture, the seal releases pore pressure and causes short term permeability enhancement

in the form of microfractures and micropores around grains. This is corroborated by fractures in the feldspar grains, while at semi-ductile conditions with quartz recrystalization.

The later stages of brittle deformation formed along the suitably oriented ductile shear zones resulting in the current fault zone geometry at the GTS shear zone (Figure 5b). The brittle deformation is bounded by the mylonitic fault cores. Due to the lack of damage outside these fault cores this system is believed to be an 'immature' fault, with minimal brittle slip.

Outside the fault cores the properties are governed by the 'frozen in' ductile structures. Inside the fault cores the properties are heterogeneously dispersed due to the macrofractures and their associated small scale microfractures, which reduce the seismic velocity. Recent borehole measurements from pump tests in the damage zone indicate that the transmisivity in the damage zone is ∼ $10^{-8}$ to $10^{-7}$ m²/s, while the host granodiorite has a transmisivity of ∼ $10^{-13}$ to $10^{-12}$ m²/s (Jalali et al., 2017).

Finally, in the Grimsel region there are more 'mature' brittle faults with a more pronounced damage zone and altered fault

core composition (Wehrens et al., 2016; Belgrano et al., 2016). These mature brittle fault cores consist of gouges, cataclasites, and fault breccias in the middle of a fractured damage zone (Figure 5c). The properties are expected to behave similar to the fault zone model of Faulkner et al. (2010), where there is an inverse relationship between low seismic velocity (i.e., elasticity) and high permeability around the fault core arises due to the extensive microfracturing in the brittle damage zone. The fault core in such a brittle fault typically has lower permeability than the surroundings due to the clay minerals in the gouge, cataclasite,

or fault breccia (e.g., Jefferies et al., 2006; Scholz, 1988; Leclère et al., 2015).

As rocks are exhumed and cooled, this system would transition from the ductile shear zone to a brittle damage zone. Thus, their mechanical properties and how fluids percolate through the entire shear zone would be highly dependent on the transient condition (depth or fault maturity) in which the fault occurs.




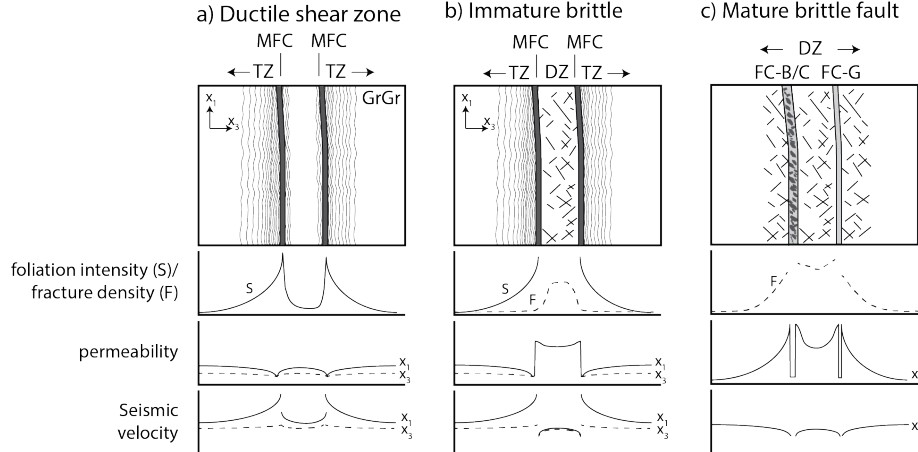

**Figure 5.** Conceptual model for the characteristics of faults in crystalline rock and their associated petrophysical properties during transition from ductile to brittle deformation conditions. a) Formation of ductile shear zones with increasing foliation in the transition zone (TZ) nearest the mylonitic fault core (MFC), b) immature brittle fault with ductile transtion zone preserved outside the MFCs and the damage zone (DZ) between, and c) mature brittle fault and DZ, where the brittle features dominate the fluid flow and elastic properties near the brittle fault cores dominated by gouge (FC-G) and breccia/cataclasite (FC-B/C) (after Faulkner et al., 2010).

### 5.3 Implication for geothermal energy production and waste disposal

In crystalline rocks the elastic, mechanical, and fluid flow properties are important characteristics for successful exploitation of natural resources. Mechanically, bulk strain can localize in the fault zone and the mechanical properties can govern earthquake rupture and fracture propagation. In terms of fluid flow, fault zones can act as both fluid conduits and barriers (e.g., Caine et al.,
1996). These can be significant in terms of building or releasing pore fluid pressures closely coupled to earthquake rupture (Sibson, 1990; Leclère et al., 2015). The elastic, mechanical, and fluid flow properties of fault zones are also directly linked to geothermal projects (Rowland and Sibson, 2004), as well as the security of long term waste storage (Barton et al., 1993; Hudson et al., 2011). With the technological advance of horizontal drilling and hydraulic fracturing the influence and interplay of mechanical and fluid flow anisotropy and heterogeneity are important when addressing stimulation in structurally complex
environments (e.g., Smart et al., 2014; Busetti et al., 2014).

The case at GTS emphasizes the interplay between properties controlled by matrix mineral and fracture-controlled properties. The shear zone at GTS serves as a proxy structure expected in a geothermal reservoir. Understanding the orientation of subsurface foliation and proximity to shear zones can assist the efficiency of energy production. The interplay of matrix and fracture flow in such systems should be considered as an additional complexity. When considering the circulation of fluids,
well placement in a geothermal injection/production system would need to address the geometry of subsurface heterogeneities. Present day hydrothermal fluids in the Grimsel region are flow in 'pipe'-like channels (i.e., not uniform across the shear zone) (Belgrano et al., 2016). Mapping such structures in crystalline basement will prove to be a challenge for successful development



of geothermal energy as a resource. Hydraulic stimulation is almost certainly required to enhance fluid flow in such crystalline systems. The importance of understanding the effect of mechanical discontinuities on hydraulic stimulation is shown with numerical models of damage propagation across mechanical layers (e.g., Smart et al., 2014). The heterogeneous and anisotropic elastic and fluid flow properties at GTS show that mechanical/elastic foliation heterogeneity must be determined, along with

stress magnitude and orientation when planning the optimal borehole placement, trajectory, and stimulation design. The low permeability in the ductile fault cores measured in this study suggest that there might be significant compartmentalization around such structures.

## 6   Conclusions

The shear zone at the GTS displays contrasting behavior in a single shear zone due to the fault evolution from ductile to brittle

deformation. The ductile history is 'frozen in' outside the mylonitic fault cores and is characterized by a transition zone of increasing foliation intensity from the host granodiorite towards the mylonitic fault cores. In the transition zone, the seismic velocity of the foliation parallel samples increases towards the fault core, while the velocity perpendicular to the foliation remains fairly constant. The permeability is also anisotropic and is lower in the samples nearest to and within the mylonitic fault core, suggesting that both permeability and seismic velocities in the transition zone are greatly influenced by the amount

and texture of phyllosilicates in the rock mass. Recent brittle deformation is bounded between the foliated fault cores, and constitutes macroscopic fractures and associated microfractures that rarely penetrate through the mylonitic fault cores.

The evolution of the system from the formation of the localized shear zones in the earliest observed ductile regime (ca. 21 Ma) and the current brittle regime follows three steps: 1) the localization of ductile deformation, 2) shearing along the rheological discontinuity causing higher foliation intensity in the granodiorite nearest to and mylonitization of the fault core,

and 3) subsequent brittle deformation along the foliated mylonitic fault cores. We hypothesize that the properties of this shear zone suggest that brittle deformation is 'immature' in the sense that the overprint has not effected the ductile transition zone.

Encountering such structures in geothermal reservoirs or waste disposal sites would prove to be challenging. The elastic, mechanical, and fluid flow heterogeneity caused by the mylonitic fault cores and their juxtaposition to a brittle damage zone would need to be considered for optimal engineering design of any reservoir usage system.

*Competing interests.*   No competing interests are present.

*Acknowledgements.*   The seismic velocity and permeability experiments were performed in the Rock Deformation Laboratory at ETH Zurich. We thank Andrea Moscariello for allowing us to use the QEMSCAN analysis at the University of Geneva. The Swiss National Science Foundation research grant NRP-70 (Exploration and characterization of deep underground reservoirs) provided funding for this project. This study is part of the Grimsel ISC project, established by the Swiss Competence Center for Energy Research - Supply of Electricity (SCCER-



SoE) with the support of the Swiss Commission for Technology and Innovation (CTI). Florian Amann, Evangelos Moulas, Valentin Gischig, Joseph Doetsch, Reza Jalali, and Hannes Krietsch are thanked for their comments and assistance during the course of this study.



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
