# Peer review of "Permeability and seismic velocity anisotropy across a ductile-brittle fault zone in crystalline rock"

_Solid Earth, 2018_

## Referee Comment (RC1) · Anonymous Referee #1 · 13 Apr 2018

This manuscript presents the results of a very interesting experimental study on the permeability of a ductile shear zone with a brittle overprint. These data will be very useful for geothermal exploration. The paper is well written and well organised. I recommend publication in Solid Earth after the following minor comments have been suitably considered.

Page 2, line 23: Are the authors just talking about boreholes that have intersected fossil ductile shear zones? If not, there are plenty of boreholes that target faults in crystalline rock in, for example, the Upper Rhine Graben. If of use, the drilling is summarised in the following recent paper:

Vidal, J., & Genter, A. (2018). Overview of naturally permeable fractured reservoirs in the central and southern Upper Rhine Graben: Insights from geothermal wells. Geothermics, 74, 57-73.

Page 2, lines 25-26 and Page 16, line 4: Rocks from a borehole are, of course, shielded from weathering. However, rocks collected at the surface for laboratory studies typically show little to no evidence of the hydrothermal alteration that often characterises rocks sampled at depth. Perhaps ductile shear zones are less altered, but brittle fault zones are often riddled with hydrothermal alteration. If the authors agree, perhaps a subtle change in wording is in order?

Page 2, line 30: There are a couple of recently published papers that discuss geothermal energy exploitation in the ductile crust. Perhaps these could be cited here?

Violay, M., Heap, M. J., Acosta, M., & Madonna, C. (2017). Porosity evolution at the brittle-ductile transition in the continental crust: Implications for deep hydro-geothermal circulation. Scientific Reports, 7(1), 7705.

Watanabe, N., Numakura, T., Sakaguchi, K., Saishu, H., Okamoto, A., Ingebritsen, S. E., & Tsuchiya, N. (2017). Potentially exploitable supercritical geothermal resources in the ductile crust. Nature Geoscience, 10(2), 140.

Figure 1: Is Figure 1d a photograph or a scan? Some clarification would help.

Page 5, lines 4-5: Is this because short samples cannot be measured in the apparatus used, or because there are problems associated with measuring the elastic wave velocities of short samples? On that note, permeability measurements on samples with a length to diameter ratio less than 1:1 are not recommended. However, I completely understand that these borehole samples are both rare and difficult to core/prepare. Perhaps the authors could indicate which samples are below 1:1 in Table 2 (I guess these are the samples with no elastic wave velocity measurements)?

Line 14: Recent laboratory experiments have shown that thermal microcracking in

granite starts at a temperature of 70 °C (see reference below). I'd recommend, in future studies, that the authors use a lower temperature to dry their samples.

Griffiths, L., Lengliné, O., Heap, M. J., Baud, P., & Schmittbuhl, J. (2018). Thermal Cracking in Westerly Granite Monitored Using Direct Wave Velocity, Coda Wave Interferometry, and Acoustic Emissions. Journal of Geophysical Research: Solid Earth.

Page 6, line 1: Impermeable means that fluids cannot pass. Please change "impermeable" to "low-permeability".

Page 6, line 3: I understand the issues associated with performing long-term permeability measurements. However, I think it would be interesting to quote the magnitude of the pressure drop used in these super low permeability measurements, to give the reader a rough idea.

Page 6, line 16: I think it's worth mentioning that the permeability measurements (at 10 MPa) were performed before the elastic wave velocity measurements (up to 260 MPa). Exposure to 260 MPa could have influenced rock microstructure.

Page 6, line 12: Is it not interesting to calculate a permeability anisotropy factor, as the authors do for their elastic wave velocity measurements? The permeability anisotropy factor could then be plotted on Figure 3.

Page 6, line 26: Are t_sample and t_rock the same?

Page 6, line 30: I guess this is the full pressure range of the apparatus? Although subtle, it might be worth clarifying this point so that people don't assume these pressures relate to borehole depths.

Page 9, lines 23 and 24: "Table", not "Tabel".

Page 10, lines 7-8: Based on comments earlier in the manuscript (that there are no microcracks) these sentences have the ability to confuse. Figure 4 shows that the elastic wave velocities increase with increasing confining pressure. So, are there microcracks? Intact granite can have a permeability as low as 10-22 m2 (see paper below). Perhaps there are microcracks in these rocks? Maybe grain-boundary microcracks that are difficult to see in thin section?

Meredith, P. G., Main, I. G., Clint, O. C., & Li, L. (2012). On the threshold of flow in a tight natural rock. Geophysical Research Letters, 39(4).

Page 10, line 18: I think this point should be made clearer in the methods section (Page 5, line 3). I would also be useful to see photographs/microscopic images of these cores to assure the reader that the comparison is reasonable. However, since this sample also contains a quartz-filled vein, for which the authors blame for the higher permeability perpendicular to foliation, perhaps it's best to remove this 23.6 sample altogether?

Figure 3: It should be noted somewhere in the discussion that temperature and fluid content can modify the measured elastic wave velocities. See papers by:

Griffiths, L., Lengliné, O., Heap, M. J., Baud, P., & Schmittbuhl, J. (2018). Thermal Cracking in Westerly Granite Monitored Using Direct Wave Velocity, Coda Wave Interferometry, and Acoustic Emissions. Journal of Geophysical Research: Solid Earth.

Nur, A., & Simmons, G. (1969). The effect of saturation on velocity in low porosity rocks. Earth and Planetary Science Letters, 7(2), 183-193.

Page 15, line 18: If of use, the above-mentioned Violay et al. (2017) paper provides data on the evolution of porosity during the high-pressure, high-temperature deformation of Westerly granite (and also some permeability measurements). There may also be useful information in:

Tullis, J., & Yund, R. A. (1977). Experimental deformation of dry Westerly granite. Journal of Geophysical Research, 82(36), 5705-5718.

Wong, T. F. (1982). Effects of temperature and pressure on failure and post-failure behavior of Westerly granite. Mechanics of Materials, 1(1), 3-17.

Page 15, line 27: Please give a reference for "...is dissimilar to models derived from brittle fault zones".

Page 15, line 33: Please give a reference for "velocities would be expected to decrease due to microfracturing".

Page 16, line 5: "losing", not "loosing".

Page 19, line 4: I agree that, due to the foliation, there's likely a large strength anisotropy. As the authors suggest, a large strength anisotropy has important ramifications for the design of engineering structures. There are a couple of papers on this topic, which may be of interest here:

Rawling, G. C., Baud, P., & Wong, T. F. (2002). Dilatancy, brittle strength, and anisotropy of foliated rocks: Experimental deformation and micromechanical modeling. Journal of Geophysical Research: Solid Earth, 107(B10).

Baud, P., Louis, L., David, C., Rawling, G. C., & Wong, T. F. (2005). Effects of bedding and foliation on mechanical anisotropy, damage evolution and failure mode. Geological Society, London, Special Publications, 245(1), 223-249.

Griffiths, L., Heap, M. J., Xu, T., Chen, C. F., & Baud, P. (2017). The influence of pore geometry and orientation on the strength and stiffness of porous rock. Journal of Structural Geology, 96, 149-160.

---

## Referee Comment (RC2) · T. Tesei (Referee) · 16 Apr 2018

Wenning et al. present a solid dataset about permeability and elastic properties of core rock materials coming from a shear zone in an underground research facility, with the aim to better understand seismic properties and potential for exploitation of such shear zones. The paper i well written, and combines some microstructural work with solid laboratory measurements of permeability and seismic wavespeeds.

My only major comment on this manuscript is with regard to the terminology used to present the data. The manuscript is presented, in particular in the abstract, as a study of brittle fault zone overprinting a ductile shear zone. Judging from the microstructures

presented in figure 2 and from the location of samples in figure 1, measurements were performed only on cores of a ductile shear zone rather than a brittle fault (i.e. on mylonites and ultramylonites). The authors often refer to ultramylonitic shear bands as "fault core" and mention the importance of inheritance of ductile structures on brittle structures. It is true that localized small scale fracturing, possibly concomitant with hydrothermal alteration and dissolution-precipitation mechanisms, may have acted during the exhumation of the shear zone in the so-called "transition zone" TZ. However the shear zone appeared to be essentially ductile, with only a late reactivation as a brittle fault in the zone named "DZ" within the ultramylonites. DZ rocks, i.e. the only clearly brittle rocks presented, were not characterized in this study, which in turn focused on essentially purely ductile shear zones. Therefore the brittle overprint, supposedly influenced by ductile structures, were not investigated. The conceptual model of the shear zone is correctly depicted by the authors in the discussion section, however I think that they should clearly term the ultramylonites "shear zones" and not fault cores, and avoid using the common terminology of brittle faults, to avoid confusion in the readership. The think that the title may be misleading in the same way. Something like "permeability and seismic velocity anisotropy of ductile shear zones enveloping a brittle fault" would be more appropriate, I reckon.

A part from this point I have only a few other minor comments on this manuscript (below) and I think this should be accepted for publication afte minor modifications.

Telemaco Tesei (Durham University)

Minor Comments P=page L=line P1, L15-16: a few references could be appropriate here. P2, L16-17: a couple of references to mechanical/geological studies of reactivation of previously ductile faults/materials:

Bolognesi, F. and Bistacchi, A., 2016. Weakness and mechanical anisotropy of phyllosilicate-rich cataclasites developed after mylonites of a low-angle normal fault (Simplon Line, Western Alps). Journal of Structural Geology, 83, pp.1-

12. Donath, F. A., 1961. Experimental study of shear failure in anisotropic rocks. Geological Society of America Bulletin, 72(6), 985-989, doi:10.1130/0016-7606(1961)72[985:ESOSFI]2.0.CO;2. P3, L1: are these two shear zones "ductile"? Or are they clay-rich fault cores/principal slip zones? P4: L13 "boundaries". P4 L 24: I would make another reference to figure 2 here. P5, L4: I would state clearly state the dimensions of the core here (or add a table, but it is impractical to read), before saying that they are not suitable for deformation experiments or seismic velocity experiments, and therefore only permeability measurements were performed on these samples. P5, L7. the samples size mentioned here suggest that for some samples the length/width ratio may be between roughly 1 and 2, contrary to previously stated P8, L2. The "void" is strange in this sentence. I would simply say "lack of " or "don't have" open microcracks. P8 L10: indicate that...

Figure 1: "Mapped" is spelled wrong in the inset of Fig. 1A. Figure 3: in the caption, mention the experimental conditions, in particular the effective confining pressure. Figure 4 caption: mention which core the measurements are taken from, and at which experimental conditions.

---

## Author Comment (AC1) · 30 Apr 2018

We would like to thank the Anonymous reviewer for thoroughly reading and commenting on our paper. The comments have improved the previous manuscript. The comments from the reviewer are included with our 'Author response.' Where appropriate we have made changes in the manuscript, which are commented and highlighted in the attached PDF (Reviewer Comment 2 are also included). For justification see the 'Author response' associated with each comment below:

Review 1

This manuscript presents the results of a very interesting experimental study on the permeability of a ductile shear zone with a brittle overprint. These data will be very useful for geothermal exploration. The paper is well written and well organised. I recommend publication in Solid Earth after the following minor comments have been suitably considered.

1. Page 2, line 23: Are the authors just talking about boreholes that have intersected fossil ductile shear zones? If not, there are plenty of boreholes that target faults in crystalline rock in, for example, the Upper Rhine Graben. If of use, the drilling is summarised in the following recent paper: Vidal, J., & Genter, A. (2018). Overview of naturally permeable fractured reservoirs in the central and southern Upper Rhine Graben: Insights from geothermal wells. Geothermics, 74, 57-73. Author response: yes the focus was meant to be ductile, we changed to read "fossil ductile shear zones" and included the following reference: "Drilling into fractured crystalline rock for geothermal exploitation has been ongoing since the 1970s [Vidal and Genter, 2018]."

2. Page 2, lines 25-26 and Page 16, line 4: Rocks from a borehole are, of course, shielded from weathering. However, rocks collected at the surface for laboratory studies typically show little to no evidence of the hydrothermal alteration that often characterises rocks sampled at depth. Perhaps ductile shear zones are less altered, but brittle fault zones are often riddled with hydrothermal alteration. If the authors agree, perhaps a subtle change in wording is in order? Author response: added: "However, precaution should be taken when assessing the extent and timing of hydrothermal alteration associated with faults at depth."

3. Page 2, line 30: There are a couple of recently published papers that discuss geothermal energy exploitation in the ductile crust. Perhaps these could be cited here? Violay, M., Heap, M. J., Acosta, M., & Madonna, C. (2017). Porosity evolution at the brittle-ductile transition in the continental crust: Implications for deep hydro-geothermal circulation. Scientific Reports, 7(1), 7705. Watanabe, N., Numakura, T., Sakaguchi, K., Saishu, H., Okamoto, A., Ingebritsen, S. E., & Tsuchiya, N. (2017). Potentially

exploitable supercritical geothermal resources in the ductile crust. Nature Geoscience, 10(2), 140. Author response: We agree and have now cited these publications.

4. Figure 1: Is Figure 1d a photograph or a scan? Some clarification would help. Author response: it's a photo, and is now described in the caption.

5. Page 5, lines 4-5: Is this because short samples cannot be measured in the apparatus used, or because there are problems associated with measuring the elastic wave velocities of short samples? On that note, permeability measurements on samples with a length to diameter ratio less than 1:1 are not recommended. However, I completely understand that these borehole samples are both rare and difficult to core/prepare. Perhaps the authors could indicate which samples are below 1:1 in Table 2 (I guess these are the samples with no elastic wave velocity measurements)?

Author response: The paragraph was reordered and reworded to make this point clearer and follow additional recommendations from 'Reviewer Comment 2:'

New paragraph: In order to determine the spatial relationship of the physical properties in the shear zone a continuous set of samples was cored every 0.1 m in the transition zone from 19.6 m to the boarder of the first MFC at 20.1 m. Abundant fractures in the damage zone between the two MFCs prevented continuous coring. Two mutually perpendicular core samples, one parallel (x1) and one perpendicular (x3) to the Grimsel granodiorite foliation were taken to characterize the physical property and anisotropy changes as a gradient away from the fault core. Sampling farther than 19.5 m was not possible due to previously made overcoring stress measurements (Figure 1c). In order to optimize the number of samples, the x1 direction was taken ~15 degrees off axis from the lineation (Figure 1d). Foliation perpendicular samples could not be taken at 19.5 and 20.1 m because of breaks in the core. The x1 and x3 samples were bored out of the core using a diamond drill bit (~2.54 cm inner diameter) with water as the cooling fluid. The 2.49 to 5.56 cm long samples were grinded and polished to craft parallel ends. To characterize the MFC, parallel and perpendicular to foliation samples were

taken at 20.2 and 23.6 m, respectively. A maximum length ($\sim$ 2.49 cm) to diameter ($\sim$ 2.53 cm) ratio of approximately 1:1 in the MFC samples, due to the extremely fissile nature of these rocks. Additionally, these two samples come from separate but similar MFC at the base of the borehole due to limited sample material. Since the seismic velocity measurements require longer samples due to signal noise and wave propagation issues, the MFC samples are only long enough to perform only permeability measurements. Additionally, two sets of perpendicular samples were taken 5 and 7 m from the start of the borehole as a background Grimsel granodiorite reference.

6. Line 14: Recent laboratory experiments have shown that thermal microcracking in granite starts at a temperature of 70 âǙęC (see reference below). I'd recommend, in future studies, that the authors use a lower temperature to dry their samples. Griffiths, L., Lengliné, O., Heap, M. J., Baud, P., & Schmittbuhl, J. (2018). Thermal Cracking in Westerly Granite Monitored Using Direct Wave Velocity, Coda Wave Inter- ferometry, and Acoustic Emissions. Journal of Geophysical Research: Solid Earth. Author response: thanks for the recommendation, we will consider this for future drying.

7. Page 6, line 1: Impermeable means that fluids cannot pass. Please change "impermeable" to "low-permeability". Author response: changed to: "the sample permeability was so low..."

8. Page 6, line 3: I understand the issues associated with performing long-term permeability measurements. However, I think it would be interesting to quote the magnitude of the pressure drop used in these super low permeability measurements, to give the reader a rough idea. Author response: For these samples, only the beginning part of the partial pressure gradient equilibration has been assessed (correlating to a pressure drop of <0.1 MPa).

9. Page 6, line 16: I think it's worth mentioning that the permeability measurements (at 10 MPa) were performed before the elastic wave velocity measurements (up to 260 MPa). Exposure to 260 MPa could have influenced rock microstructure. Author response: This would be especially important in sandstones and shales, or soft rocks that would undergo substantial plastic deformation at high confining pressures. The granodiorite rocks that we measure are elastically stiff. We have performed seismic velocity experiments under loading and unloading conditions and observe minimal seismic velocity changes in the low confining pressure experiments. Therefore, we expect the variation in permeability measurements to be within the experimental error.

10. Page 6, line 12: Is it not interesting to calculate a permeability anisotropy factor, as the authors do for their elastic wave velocity measurements? The permeability anisotropy factor could then be plotted on Figure 3. Author response: We feel that it is an interesting component, but one can see the permeability values in Table 2 are sometimes double or more in the parallel samples compared to the perpendicular samples. But this isn't the case. Thus, there would be some samples with permeability anisotropy of > 100% and some less than 10% which is not graphically aesthetic. We feel the permeability curves alone allow us to compare permeability to seismic properties.

11. Page 6, line 26: Are t_sample and t_rock the same? Author response: yes, they are both t_sample now.

12. Page 6, line 30: I guess this is the full pressure range of the apparatus? Although subtle, it might be worth clarifying this point so that people don't assume these pressures relate to borehole depths. Author response: This now reads: "Measurements were recorded across the full pressure range of 30 to 260 MPa of the apparatus"

13. Page 9, lines 23 and 24: "Table", not "Tabel". Author response: Of course, thanks!

14. Page 10, lines 7-8: Based on comments earlier in the manuscript (that there are no microcracks) these sentences have the ability to confuse. Figure 4 shows that the elastic wave velocities increase with increasing confining pressure. So, are there microcracks? Intact granite can have a permeability as low as 10-22 m2 (see paper below). Perhaps there are microcracks in these rocks? Maybe grain-boundary microcracks

that are difficult to see in thin section? Meredith, P. G., Main, I. G., Clint, O. C., & Li, L. (2012). On the threshold of flow in a tight natural rock. Geophysical Research Letters, 39(4). Author response: Correct, while they might not be visible, the velocities suggest there is compaction. The sentence was changed in the 'Results – characterization" section now reads: "In general, the samples do not have visible open microcracks, thus the porosity occurs between grain contacts (i.e., intergranular micropores)."

15. Page 10, line 18: I think this point should be made clearer in the methods section (Page 5, line 3). I would also be useful to see photographs/microscopic images of these cores to assure the reader that the comparison is reasonable. However, since this sample also contains a quartz-filled vein, for which the authors blame for the higher permeability perpendicular to foliation, perhaps it's best to remove this 23.6 sample altogether? Author response: The reworded paragraph, formerly Page 5, line 3, hopefully clarifies the difficulty in sampling these mylonitic fault cores. Especially the sentence: "Additionally, these two samples come from separate but similar MFC at the base of the borehole due to limited sample material." While sample 23.6 does not necessarily help our case, it does not hurt our case either, and we think keeping the data point with its description keeps transparency in place.

16. Figure 3: It should be noted somewhere in the discussion that temperature and fluid content can modify the measured elastic wave velocities. See papers by: Griffiths, L., Lengliné, O., Heap, M. J., Baud, P., & Schmittbuhl, J. (2018). Thermal Cracking in Westerly Granite Monitored Using Direct Wave Velocity, Coda Wave Inter- ferometry, and Acoustic Emissions. Journal of Geophysical Research: Solid Earth. Nur, A., & Simmons, G. (1969). The effect of saturation on velocity in low porosity rocks. Earth and Planetary Science Letters, 7(2), 183-193. Author response: A reference to these two works is now included in the second sentence of the second paragraph in 'Discussion – Shear zone characterization.' 17. "Additionally, temperature and fluid content can modify measured elastic wave velocities (e.g., Griffiths et al., 2018; Nur and Simmons, 1969)"

Page 15, line 18: If of use, the above-mentioned Violay et al. (2017) paper provides data on the evolution of porosity during the high-pressure, high-temperature deformation of Westerly granite (and also some permeability measurements). There may also be useful information in: Tullis, J., & Yund, R. A. (1977). Experimental deformation of dry Westerly granite. Journal of Geophysical Research, 82(36), 5705-5718. Wong, T. F. (1982). Effects of temperature and pressure on failure and post-failure behavior of Westerly granite. Mechanics of Materials, 1(1), 3-17. Author response: Reference to Violay et al., 2017 is now added with a description of the author's experimental work: "Violay et al., 2017 performed triaxial deformation experiments in Westerly granite across the brittle-ductile transition with simultaneous measurements of porosity. The authors found that the deformation in the ductile regime is associated with compaction, while the brittle regime is primarily dilatant." In addition, a sentence has been added in the introduction describing these experiments: "Violay et al., 2017 performed triaxial deformation experiments across the brittle-ductile transition in Westerly granite and show that porosity changes in the ductile regime is compactant, while the brittle regime is marked by dilation." 18. Page 15, line 27: Please give a reference for ". . .is dissimilar to models derived from brittle fault zones". Author response: Added citation to previously referenced Caine et al., 1996 and Faulkner et al., 2010. 19. Page 15, line 33: Please give a reference for "velocities would be expected to decrease due to microfracturing". Author response: added reference to: Birch, F. 1961, The velocity of compressional waves in rocks to 10 kilobars, 2. Journal of Geophysical Research. Siegesmund, S., Kern, H., Vollbrecht, A. 1991, The effect of oriented microcracks on seismic velocities in an ultramylonite. Tectonophysics.

20. Page 16, line 5: "losing", not "loosing". Author response: corrected

21. Page 19, line 4: I agree that, due to the foliation, there's likely a large strength anisotropy. As the authors suggest, a large strength anisotropy has important ramifications for the design of engineering structures. There are a couple of papers on this topic, which may be of interest here: Rawling, G. C., Baud, P., & Wong, T. F. (2002).

Dilatancy, brittle strength, and anisotropy of foliated rocks: Experimental deformation and micromechanical model- ing. Journal of Geophysical Research: Solid Earth, 107(B10). Baud, P., Louis, L., David, C., Rawling, G. C., & Wong, T. F. (2005). Effects of bedding and foliation on mechanical anisotropy, damage evolution and failure mode. Geological Society, London, Special Publications, 245(1), 223-249. Griffiths, L., Heap, M. J., Xu, T., Chen, C. F., & Baud, P. (2017). The influence of pore geometry and orientation on the strength and stiffness of porous rock. Journal of Structural Geology, 96, 149-160. Author response: We give reference to the suggested papers: "Mechanical anisotropy and heterogeneous pore geometries have been shown to have considerable influence on damage evolution and failure mode [Rawlings et al., 2002; Baud et al., 2005; Griffiths et al., 2017]."

Please also note the supplement to this comment:
https://www.solid-earth-discuss.net/se-2018-15/se-2018-15-AC1-supplement.pdf

---

## Author Comment (AC2) · 30 Apr 2018

We would like to thank Telemaco Tesei for thoroughly reading and commenting on our paper. The comments have improved the previous manuscript. The comments from the reviewer are included with our 'Author response.' Where appropriate we have made changes in the manuscript, which are commented and highlighted in the attached PDF (Reviewer Comment 1 are also included). For justification see the 'Author response' associated with each comment below:

Review 2

[Figure]

Wenning et al. present a solid dataset about permeability and elastic properties of core rock materials coming from a shear zone in an underground research facility, with the aim to better understand seismic properties and potential for exploitation of such shear zones. The paper is well written, and combines some microstructural work with solid laboratory measurements of permeability and seismic wavespeeds. 1. My only major comment on this manuscript is with regard to the terminology used to present the data. The manuscript is presented, in particular in the abstract, as a study of brittle fault zone overprinting a ductile shear zone. Judging from the microstructures presented in figure 2 and from the location of samples in figure 1, measurements were performed only on cores of a ductile shear zone rather than a brittle fault (i.e. on mylonites and ultramylonites). The authors often refer to ultramylonitic shear bands as "fault core" and mention the importance of inheritance of ductile structures on brittle structures. It is true that localized small scale fracturing, possibly concomitant with hydrothermal alteration and dissolution-precipitation mechanisms, may have acted during the exhumation of the shear zone in the so-called "transition zone" TZ. However the shear zone appeared to be essentially ductile, with only a late reactivation as a brittle fault in the zone named "DZ" within the ultramylonites. DZ rocks, i.e. the only clearly brittle rocks presented, were not characterized in this study, which in turn focused on essentially purely ductile shear zones. Therefore the brittle overprint, supposedly influenced by ductile structures, were not investigated. The conceptual model of the shear zone is correctly depicted by the authors in the discussion section, however I think that they should clearly term the ultramylonites "shear zones" and not fault cores, and avoid using the common terminology of brittle faults, to avoid confusion in the readership. The think that the title may be misleading in the same way. Something like "permeability and seismic velocity anisotropy of ductile shear zones enveloping a brittle fault" would be more appropriate, I reckon.

Author Response: We are in agreement that a change in terminology is in order. We have chosen to change the previously named mylonitic fault core (MFC) to simply mylonitic core (MC). By removing fault, we remove association to brittle deformation. We
choose to keep the term 'core' since the transition zone and mylonitic core are all part of the whole shear zone (as depicted in Figure 1c). We acknowledge the suggestion for the title change, but feel that the current title will reach a broader audience studying ductile-brittle transitions.

A part from this point I have only a few other minor comments on this manuscript (below) and I think this should be accepted for publication after minor modifications. Telemaco Tesei (Durham University)

2. Minor Comments P=page L=line P1, L15-16: a few references could be appropriate here.

Author response: The two references in the previous sentence serve as the foundation for this paragraph, and the following paragraphs describe in detail the work that has been done. We changed the references to read (see reviews: Sibson, 1994; Faulkner et al., 2010) in the sentence before as opposed to 'e.g.' in the previous sentence.

3. P2, L16-17: a couple of references to mechanical/geological studies of reactivation of previously ductile faults/materials: Bolognesi, F. and Bistacchi, A., 2016. Weakness and mechanical anisotropy of phyllosilicate-rich cataclasites developed after mylonites of a low-angle normal fault (Simplon Line, Western Alps). Journal of Structural Geology, 83, pp.1-12. Donath, F. A., 1961. Experimental study of shear failure in anisotropic rocks. Geological Society of America Bulletin, 72(6), 985-989, doi:10.1130/0016- 7606(1961)72[985:ESOSFI]2.0.CO;2.

Author response: We agree, these references has been added.

4. P3, L1: are these two shear zones "ductile"? Or are they clay-rich fault cores/principal slip zones?

Author response: They are ductile. Added "foliated ductile shear zones"

5. P4: L13 "boundaries".

Author response: corrected the spelling error.

6. P4 L 24: I would make another reference to figure 2 here.

Author response: made the reference to figure 2.

7. P5, L4: I would state clearly state the dimensions of the core here (or add a table, but it is impractical to read), before saying that they are not suitable for deformation experiments or seismic velocity experiments, and therefore only permeability measurements were performed on these samples. P5, L7. the samples size mentioned here suggest that for some samples the length/width ratio may be between roughly 1 and 2, contrary to previously stated

Author response: The paragraph was reordered and reworded to make this point clearer and follow additional recommendations from 'Reviewer Comment 1:'

New paragraph: In order to determine the spatial relationship of the physical properties in the shear zone a continuous set of samples was cored every 0.1 m in the transition zone from 19.6 m to the boarder of the first MFC at 20.1 m. Abundant fractures in the damage zone between the two MFCs prevented continuous coring. Two mutually perpendicular core samples, one parallel (x1) and one perpendicular (x3) to the Grimsel granodiorite foliation were taken to characterize the physical property and anisotropy changes as a gradient away from the fault core. Sampling farther than 19.5 m was not possible due to previously made overcoring stress measurements (Figure 1c). In order to optimize the number of samples, the x1 direction was taken ~15 degrees off axis from the lineation (Figure 1d). Foliation perpendicular samples could not be taken at 19.5 and 20.1 m because of breaks in the core. The x1 and x3 samples were bored out of the core using a diamond drill bit (~2.54 cm inner diameter) with water as the cooling fluid. The 2.49 to 5.56 cm long samples were grinded and polished to craft parallel ends. To characterize the MFC, parallel and perpendicular to foliation samples were taken at 20.2 and 23.6 m, respectively. A maximum length (~ 2.49 cm) to diameter (~ 2.53 cm) ratio of approximately 1:1 in the MFC samples, due to the extremely fissile

nature of these rocks. Additionally, these two samples come from separate but similar MFC at the base of the borehole due to limited sample material. Since the seismic velocity measurements require longer samples due to signal noise and wave propagation issues, the MFC samples are only long enough to perform only permeability measurements. Additionally, two sets of perpendicular samples were taken 5 and 7 m from the start of the borehole as a background Grimsel granodiorite reference.

8. P8, L2. The "void" is strange in this sentence. I would simply say "lack of " or "don't have" open microcracks.

Author response: changed to "… the samples do not have visible open microcracks,…"

9. P8 L10: indicate that. . .

Author response: we prefer "…, indicating hydrothermal alteration occurred."

10. Figure 1: "Mapped" is spelled wrong in the inset of Fig. 1A.

Author response: Good catch, it is now corrected.

11. Figure 3: in the caption, mention the experimental conditions, in particular the effective confining pressure.

Author response: Added the measurement conditions to the figure caption.

12. Figure 4 caption: mention which core the measurements are taken from, and at which experimental conditions.

Author response: Added that the depth corresponds to the sample name in Table 2.

Please also note the supplement to this comment:
https://www.solid-earth-discuss.net/se-2018-15/se-2018-15-AC2-supplement.pdf